# COVID-19 and Dentistry: Prevention in Dental Practice, a Literature Review

**DOI:** 10.3390/ijerph17124609

**Published:** 2020-06-26

**Authors:** Federico Alcide Villani, Riccardo Aiuto, Luigi Paglia, Dino Re

**Affiliations:** Department of Biomedical, Surgical and Dental Sciences, University of Milan, 20122 Milan, Italy; fedeberkeley@hotmail.com (F.A.V.); riccardo.aiuto@unimi.it (R.A.); dino.re@unimi.it (D.R.)

**Keywords:** dentistry, Covid-19, hygiene, infection, PPE, dental practice, mask

## Abstract

SARS-CoV-2 is a member of the family of coronaviruses. The first cases were recorded in Wuhan, China, between December 2019 and January 2020. Italy is one of the most affected countries in Europe. COVID-19 is a new challenge in modern dentistry. New guidelines are required in dental clinics to avoid contagion caused by cross-infections. A narrative review was performed using both primary sources, such as scientific articles and secondary ones, such as bibliographic indexes, web pages, and databases. The main search engines were PubMed, SciELO, and Google Scholar. Twelve articles were selected to develop the bibliographic review by applying pre-established inclusion and exclusion criteria. Precautionary measures should be applied to control COVID-19 in clinical practice. Several authors have highlighted the importance of telephone triage and/or clinic questionnaires, body temperature measurement, usage of personal protective equipment, surface disinfection with ethanol between 62% and 71%, high-speed instruments equipped with an anti-retraction system, four-handed work, and large-volume cannulas for aspiration. Clinically, the use of a rubber dam is essential. FFP2 (or N95) and FFP3 respirators, if compared to surgical masks, provide greater protection for health workers against viral respiratory infections. Further accurate studies are needed to confirm this.

## 1. Introduction

This article is a narrative review. Zoonotic diseases constitute a large group of infections that can be transmitted from animals to humans, regardless of the presence of vectors [1]. Approximately 80% of viruses, 50% of bacteria, and 40% of fungi are capable of generating a zoonotic infection [2]. Bats are considered important reservoirs and vectors for the exponential spread of zoonotic infectious diseases; they are associated with SARS and Ebola, the latter of which was responsible for an epidemic with its epicenter in Sub-Saharan Africa in 2014 [3]. SARS coronavirus in 2003 and 2019, and H1N1 flu in 2009 have demonstrated how a zoonotic infection can spread rapidly among humans, causing potentially irreversible global repercussions, from an economic, social, and health-related standpoint [2]. Compared to previous eras, globalization and the intensification of international movements have greatly facilitated the spread of viruses [1,2,3,4].

### 1.1. SARS-CoV-2: Characteristics and Mechanism of Action

Coronaviruses are a subfamily of viruses [5]. All viruses contain nucleic acids, either DNA or RNA, and a protein coat which encases the nucleic acid. Some viruses are also enclosed by an envelope of fat and protein molecules [5,6].

The SARS-Cov-2, previously named 2019-novel coronavirus by the World Health Organization (WHO), is a beta-coronavirus [5]. Coronaviruses contain an enveloped, non-segmented, positive-sense RNA genome of ~30 kb [7] with high rates of mutation and recombination [8].

Beta-coronaviruses are represented by two types of coronaviruses capable of causing severe respiratory tract infection, namely SARS-CoV and MERS-CoV [5].

The genomic sequence of SARS-CoV-2 is 96.2% similar to that of CoV-RaTG13 in bats. In contemporary literature, there are some articles that advance the hypothesis of a close correlation between SARS-CoV-2 and bats, even if the etiopathogenesis has not yet been scientifically proven [5,9]. Current studies show the high transmission capacity of the virus, with a basic reproduction number (R0) varying between 1.4 and 6.5 [10] or 2.6 and 4.7 [11].

Both measles and varicella have an R0 higher than that of COVID-19, being considerably more contagious. In the case of measles, R0 varies between 12 and 18, while for varicella it is between 10 and 12 [12,13]. SARS-CoV-2 uses the angiotensin converting enzyme, a glycoprotein localized at the endothelium of the pulmonary capillaries, as a cellular receptor for human infection [5,9]. Regarding the clinical symptomatology, COVID-19 acts directly on the upper and lower respiratory tract. The characteristic symptoms of the infection are fever, cough, general malaise, ageusia, shortness of breath, and asthenia; gastrointestinal complications have been reported, albeit rarely [14]. However, diffuse alveolar damage is the most commonly observed finding with respiratory virus infections both in the acute and late stages [15]; these medical conditions require patient hospitalization and a potential overload for the National Health Service (NHS) [16]. Comorbidities in infected patients, such as pre-existing aerial infections, heart failure, liver failure, tumors, or systemic alterations, coincide with a drastic worsening of prognosis [11,17]. The transmission of COVID-19 is variable and can occur in different circumstances: first through coughs and sneezes; second, through contact with surfaces directly exposed to the virus; and last by inhaling aerosols [11]. Initial studies have been performed to evaluate the half-life of the virus on different surfaces and for determining the severity of infection through surface contact. In a comparative study between SARS-CoV-1 and SARS-CoV-2, it has been observed that the capacity of surface stability in each virus is similar; SARS-CoV-2 is more resistant on stainless steel and plastic, and less on cardboard and copper. Although on the first two mentioned surfaces it can persist for almost 72 h, it progressively loses its viral load [18]. The most accredited transmission methods between humans are respiratory droplets and the fecal-oral route [19]. In previous coronavirus-related epidemics, specifically MERS-CoV, the risk of nosocomial infection was considerable. Therefore, the need to determine guidelines for the correct disinfection of the working health environment is the first step in controlling infections [20].

### 1.2. Epidemiology of the Coronavirus Pandemic

Towards the last week of December 2019, cases of abnormal pneumonia with unknown etiology were recorded in Wuhan, the capital of the Hubei province, in the geographical heart of the People’s Republic of China [11]. In the second half of January, the Chinese competent authorities confirmed 6000 cases of patients infected with SARS-CoV-2, although 80,000 cases were estimated at that time [21]. However, unlike SARS-CoV-1, SARS-CoV-2 has shown a greater tendency for rapid human-to-human transmission, with an R0 varying between 1.4 and 6.5, and an incubation period ranging from 2 to 14 days, with an average of 7 days [10]. On 31 January 2020, 213 deaths had been confirmed globally in 19 different countries [11]. According to the data of 14 March 2020, Italy was the most affected European country, followed by Spain [22]. On 3 May 2020, the number of people currently positive in Italy was 100,179, with 28,884 deaths [23]. The average age of people who died of COVID-19 was 78.5 years, while, the average age of diagnosis was 65 [22]. The age group with the highest mortality rate was 80 to 89 years, with a male predominance (67%). The mortality rate in the male population increased by 10% (77%) in the 70–79 age group [22]. Forty-eight percent of patients deceased from SARS-CoV-2 exhibited three or more comorbidities, two comorbidities (26%), one comorbidity (23%) and no comorbidity (1.2%). Hypertension, diabetes, and ischemic heart diseases are among the main preexisting pathologies. Only 1% of deaths from COVID-19 occurred in patients under the age of 50 years. Lombardy was the most affected region, accounting for 68% of the national cases, followed by Emilia Romagna (16.4%) and Veneto (4.3%) [24]. These data prompted authors to investigate the existence of a possible link between the exponential transmission of COVID-19 in certain Italian regions and the pollution of atmospheric particulate matter, the latter acting as a vector of the virus [25]. However, this is a spurious association because there are systematic errors that determine the lack of correlation between these two factors.

### 1.3. Dentistry and SARS-CoV-2: Clinical Aspects

According to that reported by the New York Times [26], dentistry is one of the most exposed professions to the COVID-19 contagion. It is necessary to establish a clinical protocol to be applied in the working environment to avoid new infections and progressive virus spread. In daily clinical practice, the patient’s oral fluids, material contamination, and dental unit surfaces can act as sources of contagion both for the dentist and the assistant, and for the patient himself or herself. Saliva and blood droplets that are deposited on the surfaces or aerosol inhalation generated by rotating instruments and ultrasound handpieces constitute a risk for those who occupy or will occupy those environments. Therefore, the use of disinfectants and personal protective equipment (PPE) remain essential for the proper development of the dental profession [27].

The sudden spread of SARS-CoV-2 has determined the need to modify both preventive and therapeutic protocols in dental practice. Consequently, the need to analyze the available sources in the literature to update clinical practice is crucial.

### 1.4. Aim of the Study

The aim of this narrative review is to investigate preventive measures in dental practice by assessing the operator and patient health protection during the new COVID-19 emergency by considering past experiences in terms of prevention, as the virus was only recently discovered.

Special attention is devoted to personal protection equipment, such as respirators and surgical masks, due to the major exposition of dental workers to the coronavirus.

## 2. Materials and Methods

The authors carried out a narrative review and not a systematic review, as the topic is based on a recent event, and there are still several aspects pending to be analyzed.

The process of selecting scientifically valid sources took place over five weeks, between 1 April and 4 May 2020.

The search engines used were Pubmed, Scielo, and Google Scholar. The Boolean operators used “AND” and “OR”.

The MeSH terms for the research were: “dental care”, “dentistry”, “dental offices”, “masks”, “coronavirus”, “dental equipment”, and “disinfectants”. Non-MeSH words were “SARS-CoV-2” and “PPE” The following terms were used with Boolean operators to combine searches: “Covid-19” OR “SARS-CoV-2” OR “coronavirus” AND “dental care” OR “dental office” OR “dentistry” with no limitation to the year of publication. In addition, a second search was made: “masks” OR “disinfectants” OR “PPE” OR “dental equipment” AND “Covid-19” OR “coronavirus” OR “SARS-CoV-2”.

Included in the study were bibliographic reviews, systematic reviews, meta-analyses, randomized controlled trials, cohort studies, case reports, and studies in English, Italian, Spanish, and Portuguese. The exclusion criteria were as follows: articles not related to the topic, animal studies, full-text not available, and articles in other languages. No time limits were applied during the screening phase of the scientific articles (Figure 1).

## 3. Results

Given the heterogeneous results, the selected articles were divided into two main groups according to the treated topic: SARS-CoV-2 guidelines in dentistry (Table 1) and analysis of preventive masks used for protection against SARS-CoV-2 (Table 2).

A third group, on disinfectants, was analyzed. The results obtained demonstrate compliance and homogeneity between the authors.

In studies done by Rabenau et al. [37] and Kampf et al. [38], ethanol proved to be one of the first-choice disinfectants in percentages ranging from 80 to 95% (used as a hand rub gel) [37] or 62 to 71% (used as a surface disinfectant) [38]. The coronavirus is reduced to below recording levels in a variable lapse of time between 30 and 60 s.

In the study by Rabenau et al., similar results were observed with disinfectant based on 45% iso-propanol, 30% n-propanol, and 0.2% mecetronium ethyl sulfate. Furthermore, the use of surface disinfectants such as Mikrobac Forte (containing benzalkonium chloride and laurylamine), Khorsolin FF (containing benzalkonium chloride, glutaraldehyde, and didecyldimonium chloride), and Dismozon (containing magnesium monoperphthalate) can be valid options, even if the desired effect is obtained after 30–60 min [37]. With all tested preparations, SARS-CoV-2 was inactivated to below the limit of detection, regardless of the type of organic load (0.3% albumin, 10% fetal calf serum, and 0.3% albumin with 0.3% sheep erythrocytes).

Kampf et al. in carrier tests demonstrated the disinfectant action of ethanol at 62–71% against the SARS coronavirus in 60 s, of sodium hypochlorite between 0.1–0.5% in one min, and glutaraldehyde at 2%.

In contrast, 0.04% benzalkonium chloride, 0.06% sodium hypochlorite, and 0.55% ortho-phtalaldehyde were less effective [38].

The percentages varied in the suspension tests, where ethanol (between 78 and 95%), 2-propanol (70–100%), the combination of 45% 2-propanol with 30% 1-propanol, glutardialdehyde (0.5–2.5%), formaldehyde (0.7–1%) and povidone iodine readily inactivate the coronavirus; hypochlorite is effective at a concentration greater than 0.21% [38].

## 4. Discussion

### 4.1. Preventive Measures against COVID-19 in Dental Practice

Fundamentally, the authors agree (Table 1) that it is essential to perform an accurate telephone triage, a subsequent triage in dental clinics, and a complementary questionnaire to collect as much information as possible about the patient and his or her family members, specifically regarding symptoms and movements in the previous 14 days [27,28,29,30,31]. Temperature measurement is recommended when the patient enters the dental office; if the body temperature exceeds 37.3 °C, it is suggested the treatment be postponed [30]. In patients with a cured COVID-19 infection, the American Dental Association (ADA) guidelines propose to reschedule dental treatment at least 72 h after the resolution of the symptoms, or 7 days after the appearance of initial symptoms, such as fever controlled without antipyretics and spontaneous improvement of breathing [39]. Meng et al., in a precautionary way, set the necessary recovery period to 30 days before performing non-deferrable dental care in patients who have been infected [28]. For medical-legal issues, a patient’s self-certification is also required with regard to what he/she claims during the telephone and clinical triage phase.

The ADA and the Centers for Disease Prevention and Control (CDC) recommend keeping the waiting room empty, without magazines, and avoiding the overlap of two or more appointments. If this is not possible, the minimum distance between one patient and the other must be 2 m (6 feet) in each direction.

In extreme situations, for health protection, it is reasonable to ask patients to wait in their vehicle, if possible, or nearby to the dental clinic, and advise them by telephone call or message when it is their turn [40].

As far as pediatric dentistry is concerned, persons accompanying minor age patients are asked to come to the appointment in the smallest possible number, wear a protective mask, wait in the waiting room, and not attend the patient’s treatment to avoid the risk of aerosol inhalation [27].

Further accurate studies have been carried out to demonstrate the importance of oral rinses just before dental treatment; Costa et al., in a study in 2019, highlighted how the use of chlorhexidine at 0.12% and 0.20% alters the amount of bacteria, viruses, and fungi present in the oral biofilm, reducing the risk of cross-contamination due to aerosol [29]. Since COVID-19 is sensitive to oxidation, Peng et al. proposed rinsing with 1% hydrogen peroxide or, alternatively, with 0.2% povidone-iodine [30]. This must be interpreted with caution: saliva is constantly and cyclically renewed by the salivary glands, making the virus available again.

Regardless of the type of treatment planned, healthcare professionals, especially dentists, hygienists, and dental assistants, must follow rigid protocols related to dressing and personal protective equipment. Hair caps, protective goggles, surgical masks or N95, disposable surgical gowns, special footwears, and protective visors are essential [27,28,29,30,31]. According to the “EN ISO 374-5.2016” regulation, for medical protection gloves to be considered functional against microorganisms, such as bacteria and fungi, must pass the penetration test, which analyzes air and water transition through material pores, seams, holes, and other structural imperfections [41]. “ISO 16604: 2004 method B” is an additional test that is necessary to certify the specific protection of the gloves against viruses [42].

The PPE should be used as asserted in the instructions in the user manual and must be disposed of as special waste. It is always recommended to check the integrity of the PPE, and if any negative findings, eliminate the PPE immediately [43].

### 4.2. Efficacy of Respirators and Surgical Masks against Viral Respiratory Infections

There are several articles in the scientific literature on the effectiveness of surgical masks in comparison to respirators (Table 2). The distance and length of time in which particles remain suspended in the air are determined by particle size, settling velocity, relative humidity, and air flow [36].

The European Standard classifies filtering facepiece respirators (FFP) into three categories: FFP1, FFP2, and FFP3 with minimum filtration efficiencies of 80%, 94%, and 99%. Consequently, FFP2 respirators are approximately equivalent to N95, and therefore recommended for use in the prevention of airborne infectious diseases in the US and other countries [44,45].

Both Long et al. [32] and Radonovich et al. [34], in their respective analyses did not find significant differences between the N95 and surgical masks in terms of protection from the influenza virus. Similar results were also observed in the study by Offeddu et al., which was performed two years before the current COVID-19 health emergency. On one hand, there is an equal effectiveness between the two types of masks on the influenza virus. However, compared to nonspecific respiratory tract infections, the N95 masks give slightly better results [33]. MacIntyre et al. instead obtained diametrically opposing results; they showed, through a randomized controlled clinical study on 3591 subjects, that health workers who used N95 masks continuously during the shift or in situations considered to be at high risk, presented an 85% chance of not contracting a viral infection transmitted via droplets [36].

In addition, the N95 mask group compared to the control group was associated with a significantly lower risk of contracting influenza, as confirmed by the laboratory. The authors suggest updating the classification of infectious transmissions; they consider that focusing only on aerosols and droplets is an oversimplification.

In a recent study, Ma et al. analyzed the degree of protection of surgical masks, N95, and home masks (four layers of paper and polyester) against the virus; N95 masks showed greater reliability [35].

Lee et al., focused on particles between 0.093 and 1.61 µm, and demonstrated that the FFP respirators provided better protection than the surgical masks, suggesting that such surgical masks are not a good substitute for FFP respirators in the case of airborne transmission of bacterial and viral pathogens [44]. The principal limitation of surgical masks is due to the poor face fit and the consequential possibility of aerosol aspiration [43].

In Spain, the Dentists Council (Consejo de Dentistas) reports a maximum of 4 h of use, and if kept in good condition, FFP2 or N95 masks can be sterilized through various techniques: hydrogen peroxide vapor, dry heat at 70 °C for 30 min, or in humid heat at 121 °C; however, not for more than 2–3 times [45]. The WHO protocol recommendations suggest the use of FFP3 masks according to the European nomenclature or N100, according to the United States nomenclature [46].

### 4.3. Pragmatic and Technical Recommendations during Dental Treatment in the COVID-19 Era

Hand hygiene is considered the first step in limiting the spread of the virus; WHO guidelines impose scrupulous hand-washing before and after any contact with the patient [46]. Being previously considered an essential tool for correct operating practice, the rubber dam has become even more so after the viral epidemic of 2020. Various authors underline the utility of the rubber dam on containment and protection from oral fluids; it reduces the particles present in the aerosol by 70% [30] and also drastically reduces the risk of cross-infection [27,28,30]. If it is not possible to position it, Peng et al. recommend the use of the Carisolv and an excavator for conservative treatments [30].

High-speed rotating instruments, such as the turbine and the contra-angle, must be equipped with an anti-retraction system, which prevents the release of debris and fluids that can accidentally be inhaled by healthcare professionals during clinical procedures [29,30]. Meng et al. suggests minimizing the use of these tools; if this is not possible, the last appointment of the day should be intended for those patients who need dental treatments requiring the use of high-speed rotating instruments [28]. They also recommend not to use intraoral radiographs; therefore, they propose the use of orthopantomography or CT if strictly necessary. The authors agreed on the need for four-handed work to reduce the risk of spreading the virus in the dental care unit, to manipulate the water-air syringe with extreme caution, and to use large-volume aspirators [27,28,30]. Concerning potentially deferred dental emergencies, Luzzi et al. recommend remote telephone or assistance support from the dentist. In the case of pulp pain, therapy with non-steroidal anti-inflammatory drugs, such as ibuprofen, and antibiotics, such as beta-lactams, are recommended, if the patient does not have allergies [27].

Alharbi et al. classified therapeutic dental procedures into five groups: emergencies, emergencies manageable through invasive or non-invasive procedures (minimum aerosol), non-emergencies, and elective treatments, depending on the dentist. Among the emergencies, the authors highlight maxillofacial fractures that compromise the respiratory tract, uncontrolled post-operative bleeding, and bacterial oral soft tissues infections with intra- or extra-oral swelling that negatively affect the patient’s respiratory capacity [47].

Orthodontists are suggested to stop activating the rapid palate expander; parents are instructed to reposition the Ni-Ti arch if it should go off-axis and cause a contact ulcer on the oral mucosa. Any non-urgent treatment must be postponed; if this is not possible, the dentist must follow strict protocols to avoid contagions. Peng et al., advise the elimination of waste using special yellow double-layer bags for special waste and mark them to facilitate their elimination [30].

### 4.4. Importance of Disinfectants in the Sterilization of the Dental Office

Various disinfectants available on the market, can effectively inactivate the SARS-CoV-2. The Italian Dentists Association recommends covering all surfaces, where possible, with polyethylene wrap [48]. The results obtained demonstrate compliance and homogeneity between the authors.

Rabenau et al. [37] and Kampf et al. [38] illustrated that various groups of disinfectants, such as propanol, sodium hypochlorite, and ethanol, in percentages ranging from 80 to 95% (as a hand rub) [37] or 62 to 71% (as a surface disinfectant) [38], can reduce SARS-CoV-2 load to below recording levels in a variable lapse of time. Pertinent papers on this topic are limited.

The WHO guidelines recommend the use of 5% sodium hypochlorite, with a 1:100 dilution, to be applied on surfaces for an average action time of 10 min; constant ventilation of the dental surgery room is also recommended [46]. Studies have shown that other biocidal agents such as 0.05–0.2% benzalkonium chloride or 0.02% chlorhexidine digluconate probably have lower efficiency [49].

The Spanish Dentists Council suggests the use of 1% sodium hypochlorite for the disinfection of the impressions. The action time of the disinfectant varies depending on the material used: 10 min for alginate, and 15–20 min for elastomers [45].

### 4.5. Looking toward the Future. A New Approach to the Dental Profession

As reported by Kyun-Ki et al., it is necessary to establish preventive policies in clinical and hospital settings to avoid the high risk of nosocomial infections, as with MERS [50].

Sabino-Silva et al., starting with the assumption that COVID-19 may be present in saliva through major salivary gland infection or through the crevicular fluid, suggest more accurate studies in order to evaluate the possibility of early and non-invasive virus diagnosis using saliva samples [51]. The possibility of the role of salivary gland cells in the initial progress of the infection and as a source of the virus should be considered and validated [8].

Dentistry remains one of the most exposed professions to SARS-Cov-2; each individual clinical situation must be adequately controlled and pondered by the healthcare professional; defaults in protocols cannot be tolerated. However, there are indications in the literature on how to deal with emergencies.

Currently, the swab represents the only system of diagnosis, and it requires a laboratory procedure that cannot be implemented in the dental clinic. However, rapid immunoglobulin tests, which are not considered for diagnosis, can report whether a healthcare professional has had the disease and been immunized. The development of new diagnostic tools will provide a reasonable hope for greater protection from the virus in the future. Two types of rapid tests are currently being developed for COVID-19: the first one directly detects SARS-CoV-2 antigens by nasopharyngeal secretions, while the second indirectly records the antibodies present in the serum as part of the autoimmune response against the virus [52].

Ahmed et al. conducted a cross-sectional study on 699 dental practitioners from 30 different countries using an online survey between the second and the third weeks of March 2020; 87% of participants were afraid of becoming infected with COVID-19 from either a patient or a co-worker. A considerable number of dentists (66%) wanted to close their dental cabinets until the number of COVID-19 cases declined [53].

The fear that dentists have regarding becoming infected by COVID-19 could be less if dentists and dental healthcare workers conscientiously follow the relevant recommendations [53]. Looking ahead, it is necessary to increase research efforts in aerosol control during dental treatments, including improving engineering control in dental office design. The COVID-19 pandemic has exposed important gaps in the collective response of global healthcare systems to a public health emergency [54]. Dentistry as an integral part of the health care system should be prepared to play an active role in the fight against future emerging life-threatening diseases.

## 5. Conclusions

Preventive measures against COVID-19 in dental practice include telephone and clinical triage supported by a questionnaire on recent symptoms and movements, body temperature measurement, oral rinses with 1% hydrogen peroxide, and the use of specific PPEs.

Pragmatic and technical recommendations for correct clinical practice are the implementation of anti-retraction dental handpieces, four-handed work, the use of a rubber dam, and large-volume cannulas for aspiration.

FFP2 (or N95) and FFP3 respirators, if compared to surgical masks, provide greater protection to health workers against viral respiratory infections.

Ethanol between 62% and 71% and sodium hypochlorite between 0.1% and 0.5% are considered to be the best among the surface disinfectants.

This narrative review has some limitations. As there is a current emergency, in the literature there is a limited and heterogenous number of primary sources directly related to the repercussion of SARS-CoV-2 on the dental discipline. Further studies are needed in the future.

## Figures and Tables

**Figure 1 ijerph-17-04609-f001:**
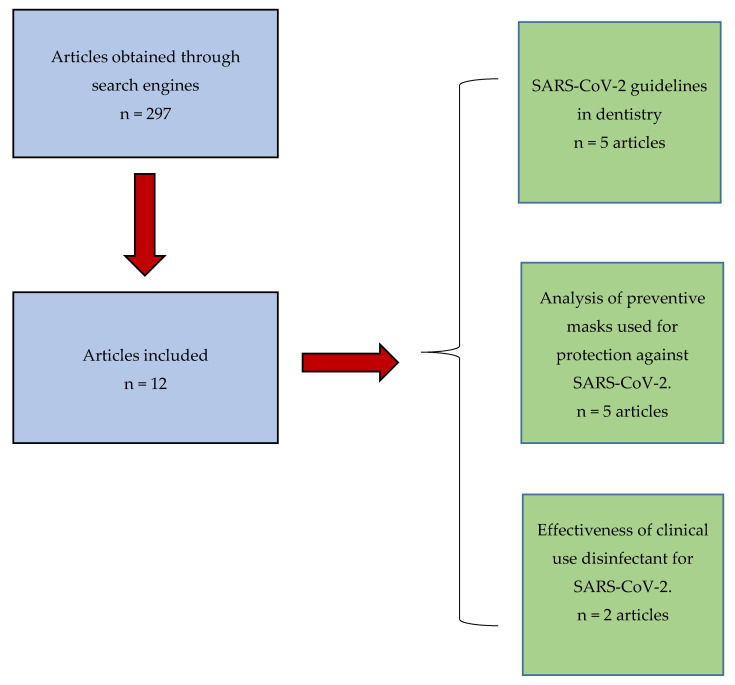
Flowchart.

**Table 1 ijerph-17-04609-t001:** SARS-CoV-2 guidelines in dentistry.

Authors/Year	Telephone Triage Questionnaire	Body Tº Measurement	Oral Rinses	PPE Hand Hygiene	Dental Handpiece	Rubber Dam	Relevant Clinical Aspects
**Meng L et al. [28]** **Y: 2020**	YES	YES	YES	Mandatory	Avoid	YES	No Intraoral X-ray
**Costa V et al. [29]** **Y: 2019**	YES	NR	YES CLX 0.12–0.2%	Mandatory	NR	NR	Aerosol Control
**Peng X et al. [30]** **Y: 2020**	YES	YES Tº > 37.3 NO tmt	YES Hyd perox 1% Povidon–Iodine 0.2%	Mandatory	YES Anti-retraction	YES	Medical waste management
**Luzzi V et al. [27]** **Y: 2020**	YES	YES	YES NO CLX	Mandatory	YES Anti-retraction	YES	High volume aspirators
**Yang Y et al. [31]** **Y: 2020**	YES	YES	YES	Mandatory	NR	NR	Operating room disinfection

—Hyd perox: hydrogen peroxide; —NR: not reported by the authors in the article; —tmt: treatment.

**Table 2 ijerph-17-04609-t002:** Analysis of preventive masks used for protection against SARS-CoV-2.

Authors/Year	Type of Study	Sample	Exposure	Masks Analyzed	Efficacy: Significant Differences
**Long Y et al. [32]** **Y: 2020**	Systematic review and meta-analysis	6 randomized controlled clinical trials	Influenza virus	N95 Surgical masks	NO
**Offeddu V et al. [33]** **Y: 2017**	Systematic review and meta-analysis	23 observational studies 6 controlled randomized trials	-Influenza virus -Non-specific respiratory infection -SARS	N95 Surgical masks	NO virus influenza YES Clinical respiratory infection (> N95)
**Radonovich LJ et al. [34].** **Y: 2019**	Randomized clinical trial	2862 healthy workers	Influenza virus	N95 Surgical masks	NO
**Ma QX et al. [35].** **Y: 2020**	Pilot study	3 typologies of masks	Avian influenza	N95 Surgical masks Homemade masks (4 layer of paper + polyester)	YES -N95: 99.98% -Mas. Surg: 97.14% -Home Masks: 95% PROTECTION
**MacIntyre et al. [36].** **Y: 2017**	Randomized controlled clinical trial	3591 healthy subjects	Influenza A Influenza B	N95 Surgical masks Control Group	YES > protection with N95

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
