# Peer review of "COVID-19 and Dentistry: Prevention in Dental Practice, a Literature Review"

_ijerph, 2020, doi:10.3390/ijerph17124609_

Round 1

Reviewer 1 Report

In the manuscript titled “Hygiene, disinfection and proper use of PPE to prevent the transmission of the new coronavirus (Covid-19) in dental practice: A literature review.”, the authors present the useful information for this current outbreak of COVID-19. More structure arrangements and language check are suggested before publication.

I recommend the major revision of this paper, to make the author address and solve the following issues:

  1. The article is difficult to read. The title is “use of PPE’ to prevent COVID-19 in ‘dental practice’, but only table 1 is about these, others are for other precautionary methods, such as wearing mask or using disinfection. I do suggest the change of the title, or rearrange the contents.
  2. Please remove the words in abstract ‘Background, Methods, Results, Conclusions’ in abstract.
  3. English Language needs to be improved. For example, why ‘but’ is used in line 35?
  4. Introduction is a bit long, and descriptions in the results are too limited. Sub sections with subtitles are suggested to put in the results part.
  5. As for the table 1, can you draw carefully, the arrow is shown on the box in line 162.
  6. As for the table 3, where are the references for the time used to inactive the virus?
  7. The discussion part is too long, separated sections with subtitle is needed.

In conclusion, this paper is interesting, and useful. The contents need to be rearranged. I recommend the publication in IJERPH after a major revision.

Author Response

Dear Editor,

Dear Reviewers,

Thank you for the attention you have paid to our manuscript and for the important suggestions that made possible to improve it. As kindly requested, we report below all the changes we made.

REVIEWER 1

  1. a) The article is difficult to read. The title is “use of PPE’ to prevent COVID-19 in ‘dental practice’, but only table 1 is about these, others are for other precautionary methods, such as wearing mask or using disinfection. I do suggest the change of the title, or rearrange the contents.

Right from the beginning, we decided to collect information about the specific preventive measures in dental office during Covid-19 epidemic. The main goal was to give an initial generic standpoint about the correlation between dentistry and SARS-CoV2-2, based on few articles available in the literature at that time. As suggested, we changed the title. The new title and the new aim of the study should help the reader to understand better what we would like to transmit through our article.

  1. b) Please remove the words in abstract ‘Background, Methods, Results, Conclusions’ in abstract.

We removed words “background”, “methods”, “results”, “conclusions” from the abstract because they were not necessary.

  1. c) English Language needs to be improved. For example, why ‘but’ is used in line 35?

We deeply improved english language; we corrected all grammar mistakes.

  1. d) Introduction is a bit long, and descriptions in the results are too limited. Sub sections with subtitles are suggested to put in the results part.

Superfluous information have been removed in the introduction part. Regarding the result part, new paragraphs have been written; this section is now more complete and thorough.

  1. e) As for the table 1, can you draw carefully, the arrow is shown on the box in line 162.

The arrow has been now drawn carefully in table 1. “Design” mistakes have been corrected.

  1. f) As for the table 3, where are the references for the time used to inactive the virus?

We eliminated table 3 because it was misleading and vague, as indicated also by Reviewer 3. All information and data exposed in table 3, have been directly described in the result part.

  1. g) The discussion part is too long, separated sections with subtitle is needed.

Discussion has been separated in section with subtitles. This makes the article easier to read and the discussion more orientated and contextualized.

Reviewer 2 Report

Providing a summary of the best practices for COVID-19 prevention in dental offices is a significant topic area, and a thorough review of published work could provide an easy summary to dental offices for the best ways to prevent disease spread. This paper; however, does not clearly summarize these findings. A more descriptive discussion of previous findings would improve the paper. In addition, more clearly stating conclusions would improve the usefulness of the paper.

  • The introduction is not focused and needs editing for clarity
  • Page 2, lines 53-54: Viruses do not contain a nucleus, their nucleic acid is not enclosed in in a compartment, referring to the nucleic acid as a nucleus is misleading
  • The first few paragraphs of the materials and methods are not necessary
  • The exclusion criteria seems too extensive if only 12 of 311 papers were used
  • While I can infer what the purpose and intended conclusions are, it is not clear based on the writing. The conclusion also doesn’t provide an overall description of what was found.

Author Response

Dear Editor,

Dear Reviewers,

Thank you for the attention you have paid to our manuscript and for the important suggestions that made possible to improve it. As kindly requested, we report below all the changes we made.

REVIEWER 2

  1. a) The introduction is not focused and needs editing for clarity.

We changed the title of the article and to make the purpose of the paper more comprehensible. After the changes we made, the different parts of the introduction should be more explanatory.  Epidemiological data can help to understand the magnitude of this pandemic. Also some superfluous paragraphs of the introduction have been removed.

  1. b) Page 2, lines 53-54: Viruses do not contain a nucleus, their nucleic acid is not enclosed in in a compartment, referring to the nucleic acid as a nucleus is misleading.

Corrections have been made. 

  1. c) The first few paragraphs of the materials and methods are not necessary. The exclusion criteria seems too extensive if only 12 of 311 papers were used.

As the Reviewer suggested, first paragraphs of material and methods may be not necessary. We removed them.

We have clarified the inclusion and exclusion criteria to justify the presence of a few articles. Unfortunately many articles were not in English and many were not relevant.

  1. d) While I can infer what the purpose and intended conclusions are, it is not clear based on the writing. The conclusion also doesn’t provide an overall description of what was found.

The new structure of the article may help the reader to understand the purpose of the study and consequently the conclusions. Anyway, new conclusion have been written again, in order to reflect exactly what is reported in the paper.

Reviewer 3 Report

The impact of COVID-19 on health care is certainly significant.  Attention to dentistry and appropriate procedures to reduce transmission in dental practice is important.  

However, your paper does not contribute substantively in this important area.  The aim is vague, being to "investigate on the current knowledge for the organization of the dental practice, by looking at the operators and patient health protection during the new emergency from Covid-19."  

The introduction is long and rambling, and includes a very large proportion that is superfluous to the paper's purpose.  Virtually all of lines 27 to 113 could be deleted.

The presentation of results is confusing.  Table 1 presents, one assumes, expert opinion on various measures that might be used in dentistry to reduce transmission (e.g. telephone triage).  There is no truly critical assessment of the efficacy of these measures.  Table 2 is entirely based on studies of viruses other than SARS-CoV-2.  Table 3 appears to be based on two papers.  It is unclear how efficacy against the virus was determined, and the results are presented as dichotomous.

The discussion is very long.  The structure of the discussion is unclear.  It includes material inappropriate to the discussion section (e.g. "In a recent study, Ma et al. analyzed the protection degree of surgical masks, N95 and home masks (4 layers of paper and polyester) against the virus; N95 masks have shown greater reliability, with a blocking action on the virus of 99.98%, followed by surgical ones with 97%", lines 263-5).

Author Response

Dear Editor,

Dear Reviewers,

Thank you for the attention you have paid to our manuscript and for the important suggestions that made possible to improve it. As kindly requested, we report below all the changes we made.

REVIEWER 3

a) Your paper does not contribute substantively in this important area.  The aim is vague, being to "investigate on the current knowledge for the organization of the dental practice, by looking at the operators and patient health protection during the new emergency from Covid-19."  

We made changes in order that the article should be now clearer and easier to read.

b) The introduction is long and rambling, and includes a very large proportion that is superfluous to the paper's purpose.  Virtually all of lines 27 to 113 could be deleted.

As suggested by Reviewers, modifications in the introduction part were made. We removed paragraphs that were superfluous and not inherent with the treated topic.

c) The presentation of results is confusing.  Table 1 presents, one assumes, expert opinion on various measures that might be used in dentistry to reduce transmission (e.g. telephone triage).  There is no truly critical assessment of the efficacy of these measures.  Table 2 is entirely based on studies of viruses other than SARS-CoV-2.  Table 3 appears to be based on two papers.  It is unclear how efficacy against the virus was determined, and the results are presented as dichotomous.

Results section has been renovated. Table 3 was removed because it was misleading; limited papers on this topic are available. Anyway, data collected by authors are described in the result.

d) The discussion is very long.  The structure of the discussion is unclear.  It includes material inappropriate to the discussion section (e.g. "In a recent study, Ma et al. analyzed the protection degree of surgical masks, N95 and home masks (4 layers of paper and polyester) against the virus; N95 masks have shown greater reliability, with a blocking action on the virus of 99.98%, followed by surgical ones with 97%", lines 263-5).

The discussion, as indicated by Reviewers, was long and not clear. We decided to create sub section and make it more understandable and organized. Material inappropriate for the discussion part has been removed.

Round 2

Reviewer 1 Report

In the abstract, please address the dental practice.

Please check the spelling and format again. For example:

  1. why there are full stop in the articel title.
  2. Line 190, '30%n-proponal', '0,2%'.
  3. Why there are still yellow markers in Figure 1.
  4. and so on...

Author Response

Dear Editor,

Dear Reviewers,

Thank you for the attention you have paid to our manuscript and for the important suggestions that made possible to improve it. As kindly requested, we report below all the changes we made.

Please check the spelling and format again. For example:

1) why there are full stop in the article title.

Full stops have been removed from the title. New title is available.

2 )Line 190, '30%n-proponal', '0,2%'.

We provided to correct the error.

3) Why there are still yellow markers in Figure 1.

Yellow markers have been removed from the flow chart.

4) and so on...

The paper has been edited to ensure that the language is clear and free of errors. The edit was performed by professional editors. We attach a certificate of english editing.

Reviewer 2 Report

There are still writing errors that need clarifying, including the newly added sentence at the end of the introduction that begins "accurate awareness..."

Author Response

Dear Editor,

Dear Reviewers,

Thank you for the attention you have paid to our manuscript and for the important suggestions that made possible to improve it. As kindly requested, we report below all the changes we made.

1) There are still writing errors that need clarifying, including the newly added sentence at the end of the introduction that begins "accurate awareness..."

The paper has been edited to ensure that the language is clear and free of errors. The edit was performed by professional editors. We attach a certificate of english editing.
